# Higher Order Priors for Joint Intrinsic Image, Objects, and Attributes Estimation

**Vibhav Vineet**
Oxford Brookes University, UK
vibhav.vineet@gmail.com

**Carsten Rother**
TU Dresden, Germany
carsten.rother@tu-dresden.de

**Philip H.S. Torr**
University of Oxford, UK
philip.torr@eng.ox.ac.uk

## Abstract

Many methods have been proposed to solve the problems of recovering *intrinsic scene properties* such as shape, reflectance and illumination from a single image, and *object class segmentation* separately. While these two problems are mutually informative, in the past not many papers have addressed this topic. In this work we explore such joint estimation of intrinsic scene properties recovered from an image, together with the estimation of the objects and attributes present in the scene. In this way, our unified framework is able to capture the correlations between intrinsic properties (*reflectance, shape, illumination*), objects (*table, tv-monitor*), and materials (*wooden, plastic*) in a given scene. For example, our model is able to enforce the condition that if a set of pixels take same object label, e.g. *table*, most likely those pixels would receive similar reflectance values. We cast the problem in an energy minimization framework and demonstrate the qualitative and quantitative improvement in the overall accuracy on the NYU and Pascal datasets.

## 1  Introduction

Recovering scene properties (shape, illumination, reflectance) that led to the generation of an image has been one of the fundamental problems in computer vision. Barrow and Tenebaum [13] posed this problem as representing each scene properties with its distinct "intrinsic" images. Over the years, many decomposition methods have been proposed [5, 16, 17], but most of them focussed on recovering a reflectance image and a shading[1] image without explicitly modelling illumination or shape. But in the recent years a breakthrough in the research on intrinsic images came with the works of Barron and Malik [1-4] who presented an algorithm that jointly estimated the reflectance, *the illumination and the shape*. They formulate this decomposition problem as an energy minimization problem that captures prior information about the structure of the world.

Further, recognition of objects and their material attributes is central to our understanding of the world. A great deal of work has been devoted to estimating the objects and their attributes in the scene: Shotton et.al. [22] and Ladicky et.al. [9] propose approaches to estimate the object labels at the pixel level. Separately, Adelson [20], Farhadi et.al. [6], Lazebnik et.al. [23] define and estimate the attributes at the pixel, object and scene levels. Some of these attributes are material properties such as *woollen, metallic, shiny*, and some are structural properties such as *rectangular, spherical*.

While these methods for estimating the intrinsic images, objects and attributes have separately been successful in generating good results on laboratory and real-world datasets, they fail to capture the strong correlation existing between these properties. Knowledge about the objects and attributes in the image can provide strong prior information about the intrinsic properties. For example, if a set of pixels takes the same object label, e.g. *table*, most likely those pixels would receive similar reflectance values. Thus recovering the objects and their attributes can help reduce the ambiguities present in the world leading to better estimation of the reflectance and other intrinsic properties.

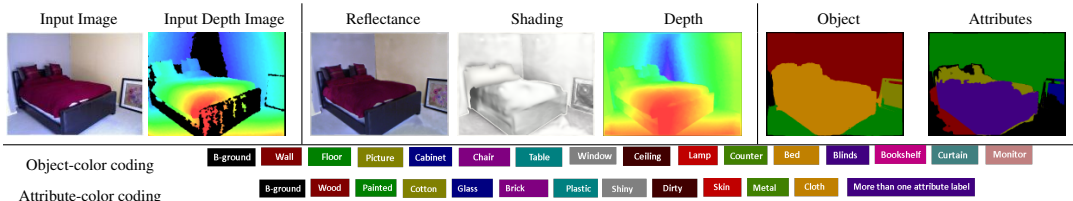

| Input Image | Input Depth Image | Reflectance | Shading | Depth | Object | Attributes |

Object-color coding: B-ground | Wall | Floor | Picture | Cabinet | Chair | Table | Window | Ceiling | Lamp | Counter | Bed | Blinds | Bookshelf | Curtain | Monitor

Attribute-color coding: B-ground | Wood | Painted | Cotton | Glass | Brick | Plastic | Shiny | Dirty | Skin | Metal | Cloth | More than one attribute label

Figure 1: *Given a RGBD image, our algorithm jointly estimates the intrinsic properties such as reflectance, shading and depth maps, along with the per-pixel object and attribute labels.*

Additionally such a decomposition might be useful for per-pixel object and attribute segmentation tasks. For example, using reflectance (illumination invariant) should improve the results-when estimating per-pixel object and attribute labels [24]. Moreover if a set of pixels have similar reflectance values, they are more likely to have the same object and attribute class.

Some of the previous research has looked at the correlation of objects and intrinsic properties by propagating results from one step to the next. Osadchy et.al. [18] use specular highlights to improve recognition of transparent, shiny objects. Liu et.al. [15] recognize material categories utilizing the correlation between the materials and their reflectance properties (e.g. *glass is often translucent*). Weijer et.al. [14] use knowledge of the objects present in the scene to better separate the illumination from the reflectance images. However, the problem with these approaches is that the errors in one step can propagate to the next steps with no possibility of recovery. Joint estimation of the intrinsic images, objects and attributes can be used to overcome these issues. For instance, in the context of joint object recognition and depth estimation such positive synergy effects have been shown in e.g. [8].

In this work, our main contribution is to explore such synergy effects existing between the intrinsic properties, objects and material attributes present in a scene (see Fig. 1). Given an image, our algorithm jointly estimates the intrinsic properties such as reflectance, shading and depth maps, along with per-pixel object and attribute labels. We formulate it in a global energy minimization framework, and thus our model is able to enforce the consistency among these terms. Finally, we use an approximate dual decomposition based strategy to efficiently perform inference in the joint model consisting of both the continuous (reflectance, shape and illumination) and discrete (objects and attributes) variables. We demonstrate the potential of our approach on the aNYU and aPascal datasets, which are extended versions of the NYU [25] and Pascal [26] datasets with per-pixel attribute labels. We evaluate both the qualitative and quantitative improvements for the object and attribute labelling, and qualitative improvement for the intrinsic images estimation.

We introduce the problem in Sec. 2. Section 3 provides details about our joint model, section 4 describes our inference and learning, Sec. 5 and 6 provide experimentation and discussion.

## 2    Problem Formulation

Our goal is to jointly estimate the intrinsic properties of the image, i.e. reflectance, shape and illumination, along with estimating the objects and attributes at the pixel level, given an image array $\bar{C} = (\bar{C}_1...\bar{C}_V)$ where $\bar{C}_i \in \mathbb{R}^3$ is the $i^{th}$ pixel's associated RGB value in the image with $i \in \mathcal{V} = \{1...V\}$. Before going into the details of the joint formulation, we consider the formulations for independently solving these problems. We first briefly describe the SIRFS (shape, illumination and reflectance from shading) model [2] for estimating the intrinsic properties for a single given object, and then a CRF model for estimating objects, and attributes [12].

### 2.1    SIRFS model for a single, given object mask

We build on the SIRFS model [2] for estimating the intrinsic properties of an image. They formulate the problem of recovering the shape, illumination and reflectance as an energy minimization problem given an image. Let $R = (R_1...R_V)$, $Z = (Z_1...Z_V)$ be the reflectance, and depth maps respectively, where $R_i \in \mathbb{R}^3$ and $Z_i \in \mathbb{R}^3$, and the illumination $L$ be a 27-dimensional vector of spherical harmonics [10]. Further, let $S(Z, L)$ be a function that generates a shading image given the depth map $Z$ and the illumination $L$. Here $S_i \in \mathbb{R}^3$ and subsumes all light-dependent properties, e.g. shadows, inter-reflections (refer to [2] for details). The SIRFS model then minimizes the energy

$$\begin{aligned} \text{minimize}_{R,Z,L} \ E^{\text{SIRFS}} &= E^R(R) + E^Z(Z) + E^L(L) \\ \text{subject to} \quad & \bar{C} = R \cdot S(Z, L) \end{aligned} \tag{1}$$

where "·" represents componentwise multiplication, and $E^R(R), E^Z(Z)$ and $E^L(L)$ are the costs for the reflectance, depth and illumination respectively. The most likely solution is then estimated by using a multi-scale L-BFGS, a limited-memory approximation of the Broyden-Fletcher-Goldfarb-Shanno algorithm [2], strategy which in practice finds better local optima than other gradient descent strategies. The SIRFS model is limited to estimating the intrinsic properties for a single object mask within an image. The recently proposed Scene-SIRFS model [4] proposes an approach to recover the intrinsic properties of whole image by embedding a mixture of shapes in a soft segmentation of the scene. In Sec. 3 we will also extend the SIRFS model to handle multiple objects. The main difference to Scene-SIRFS is that we perform joint optimization over the object (and attributes) labelling and intrinsic image properties per-pixel.

## 2.2 Multilabel Object and Attribute Model

The problem of estimating the per-pixel objects and attributes labels can also be formulated in a CRF framework [12]. Let $O = (O_1...O_V)$ and $A = (A_1...A_V)$ be the object and attribute variables associated with all $V$ pixels, where each object variable $O_i$ takes one out of $K$ discrete labels such as table, monitor, or floor. Each attribute variable $A_i$ takes a label from the power set of the $M$ attribute labels, for example the subset of attribute labels can be $A_i = \{red, shiny, wet\}$. Efficient inference is performed by first representing each attributes subset $A_i$ by $M$ binary attribute variables $A_i^m \in \{0, 1\}$, meaning that $A_i^m = 1$ if the $i^{th}$ pixel takes the $m^{th}$ attribute and it is absent when $A_i^m = 0$. Under this assumption, the most likely solution for the objects and the attributes correspond to minimizing the following energy function

$$E^{OA}(O, A) = \sum_{i \in \mathcal{V}} \psi_i(O_i) + \sum_m \sum_{i \in \mathcal{V}} \psi_{i,m}(A_i^m) + \sum_{i<j \in \mathcal{V}} \psi_{ij}(O_i, O_j) + \sum_m \sum_{i<j \in \mathcal{V}} \psi_{ij}(A_i^m, A_j^m) \quad (2)$$

Here $\psi_i(O_i)$ and $\psi_{i,m}(A_i^m)$ are the object and per-binary attribute dependent unary terms respectively. Similarly, $\psi_{ij}(O_i, O_j)$ and $\psi_{ij}(A_i^m, A_j^m)$ are the pairwise terms defined over the object and per-binary attribute variables. Finally the best configuration for the object and attributes are estimated using a mean-field based inference approach [12]. Further details about the form of the unary, pairwise terms and the inference approach are described in our technical report [29].

## 3 Joint Model for Intrinsic Images, Objects and Attributes

Now, we provide the details of our formulation for jointly estimating the intrinsic images $(R, Z, L)$ along with the objects $(O)$ and attribute $(A)$ properties given an image $\bar{C}$ in a probabilistic framework. We define the posterior probability and the corresponding joint energy function $E$ as:

$$P(R, Z, L, O, A|I) = 1/Z(I) \exp\{-E(R, Z, L, O, A|I)\}$$
$$E(R, Z, L, O, A|I) = E^{\text{SIRFSG}}(R, Z, L|O, A) + E^{RO}(R, O) + E^{RA}(R, A) + E^{OA}(O, A)$$
$$\text{subject to} \quad \bar{C} = R \cdot S(Z, L) \quad (3)$$

We define $E^{\text{SIRFSG}} = E^R(R) + E^Z(Z) + E^L(L)$, a new global energy term. The terms $E^{RO}(R, O)$ and $E^{RA}(R, A)$ capture correlations between the reflectance, objects and/or attribute labels assigned to the pixels. These terms take the form of higher order potentials defined on the image segments or regions of pixels generated using unsupervised segmentation approach of Felzenswalb and Huttenlocker [21]. Let $\mathcal{S}$ corresponds to the set of these image segments. These terms are described in detail below.

### 3.1 SIRFS model for a scene

Given this representation of the scene, we model the scene specific $E^{\text{SIRFSG}}$ by a mixture of reflectance, and depth terms embedded into the segmentation of the image and an illumination term as:

$$E^{\text{SIRFSG}}(R, Z, L|O, A) = \sum_{c \in \mathcal{S}} \left( E^R(R_c) + E^Z(Z_c) \right) + E^L(L) \quad (4)$$

where $R = \{R_c\}$, $Z = \{Z_c\}$. Here $E^R(R_c)$ and $E^Z(Z_c)$ are the reflectance and depth terms respectively defined over segments $c \in \mathcal{S}$. In the current formulation, we have assumed that we have a single model of illumination $L$ for whole scene which corresponds to a 27-dimensional vector of spherical harmonics [2].

## 3.2 Reflectance, Objects term

The joint reflectance-object energy term $E^{RO}(R, O)$ captures the relations between the objects present in the scene and their reflectance properties. Our higher order term takes following form:

$$E^{RO}(R, O) = \sum_{c \in \mathcal{S}} \pi_o^c \psi(R_c) + \sum_{c \in \mathcal{S}} \pi_r^c \psi(O_c) \tag{5}$$

where $R_c, O_c$ are the labeling for the subset of pixels $c$ respectively. Here $\pi_o^c \psi(R_c)$ is an object dependent quality sensitive higher order cost defined over the reflectance variables, and $\pi_r^c \psi(O_c)$ is a reflectance dependent quality sensitive higher order cost defined over the object variables. The term $\psi(R_c)$ reduces the variance of the reflectance values within a clique and takes the form $\psi(R_c) = \|c\|^{\theta_\alpha} (\theta_p + \theta_v G^r(c))$ where

$$G^r(c) = \exp\left(-\theta_\beta \frac{\|\sum_{i \in c}(R_i - \mu_c)^2\|}{\|c\|}\right). \tag{6}$$

Here $\|c\|$ is the size of the clique, $\mu_c = \frac{\sum_{i \in c} R_i}{\|c\|}$ and $\theta_\alpha, \theta_p, \theta_v, \theta_\beta$ are constants. Further in order to measure the quality of the reflectance assignment to the segment, we weight the higher order cost $\psi(R_c)$ with an object dependent $\pi_o^c$ that measures the quality of the segment. In our case, $\pi_o^c$ takes following form:

$$\pi_o^c = \begin{cases} 1 & \text{if } O_i = l, \ \forall i \in c \\ \lambda^o & \text{otherwise} \end{cases} \tag{7}$$

where $\lambda^o < 1$ is a constant. This term allows variables within a segment to take different reflectance values if the pixels in that segment take different object labels. Currently the term $\pi_o^c$ gives rise to a hard constraint on the penalty but can be extended to one that penalizes the cost softly as in [29].

Similarly we enforce higher order consistency over the object labeling in a clique $c \in \mathcal{S}$. The term $\psi(O_c)$ takes the form of pattern-based $P^N$-Potts model [7] as:

$$\psi(O_c) = \begin{cases} \gamma_l^o & \text{if } O_i = l, \ \forall i \in c \\ \gamma_{max}^o & \text{otherwise} \end{cases} \tag{8}$$

where $\gamma_l^o, \gamma_{max}^o$ are constants. Further we weight this term with a reflectance dependent quality sensitive term $\pi_r^c$. In our experiment we measure this term based on the variance of reflectance terms on all constituent pixels of a segment, i.e., $G^r(c)$ (define earlier). Thus $\pi_r^c$ takes following form:

$$\pi_r^c = \begin{cases} 1 & \text{if } G^r(c) < K, \ \forall i \in c \\ \lambda^r & \text{otherwise} \end{cases} \tag{9}$$

where $K$ and $\lambda^r < 1$ are constants. Essentially, this quality measurement allows the pixels within a segment to take different object labels, if the variation in the reflectance terms within the segment is above a threshold. To summarize, these two higher order terms enforce the cost of inconsistency within the object and reflectance labels.

## 3.3 Reflectance, Attributes term

Similarly we define the term $E^{RA}(R, A)$ which enforces a higher order consistency between reflectance and attribute variables. Such higher order consistency takes the following form:

$$E^{RA}(R, A) = \sum_m \left(\sum_{c \in \mathcal{S}} \pi_{a,m}^c \psi(R_c) + \sum_{c \in \mathcal{S}} \pi_r^c \psi(A_c^m)\right) \tag{10}$$

where $\pi_{a,m}^c \psi(R_c)$ and $\pi_r^c \psi(A_c^m)$ are the higher order terms defined over the reflectance image and the attribute image corresponding to the $m^{th}$ attribute respectively. Forms of these terms are similar to the one defined for the object-reflectance higher order terms; these terms are further explained in the supplementary material.

## 4 Inference and Learning

Given the above model, our optimization problem involves solving following joint energy function to get the most likely solution for $(R, Z, L, O, A)$:

$$E(R, Z, L, O, A|I) = E^{\text{SIRFSG}}(R, Z, L) + E^{RO}(R, O) + E^{RA}(R, A) + E^{OA}(O, A) \tag{11}$$

However, this problem is very challenging since it consists of both the continuous variables $(R, Z, L)$ and discrete variables $(O, A)$. Thus in order to minimize the function efficiently without losing accuracy we follow an approximate dual decomposition strategy [28].

We first introduce a set of duplicate variables for the reflectance $(R^1, R^2, R^3)$, objects $(O^1, O^2)$, and attributes $(A^1, A^2)$ and a set of new equality constraints to enforce the consistency on these duplicate variables. Our optimization problem thus takes the following form:

$$\underset{R^1, R^2, R^3, Z, L, O^1, O^2}{\text{minimize}} \quad E(R^1, Z, L) + E(O^1, A^1) + E(R^2, O^2) + E(R^3, A^2)$$

$$\text{subject to} \quad R^1 = R^2 = R^3; \quad O^1 = O^2; \quad A^1 = A^2 \qquad (12)$$

From now on we have removed the subscripts and superscripts from the energy terms for simplicity of the notations. Now we formulate it as an unconstrained optimization problem by introducing a set of lagrange multipliers $\theta_r^1, \theta_r^2, \theta_o, \theta_a$ and decompose the dual problem into four sub-problems as:

$$
\begin{aligned}
E(R^1, Z, L) \quad & + \quad E(O^1, A^1) + E(R^2, O^2) + E(R^3, A^2) + \theta_r^1(R^1 - R^2) \\
& + \quad \theta_r^2(R^2 - R^3) + \theta_o(O^1 - O^2) + \theta_a(A^1 - A^2) \\
& = \quad g_1(R^1, Z, L) + g^2(O^1, A^1) + g_3(O^2, R^2) + g_4(A^2, R^3), \qquad (13)
\end{aligned}
$$

where

$$
\begin{aligned}
g_1(R^1, Z, L) & = \text{minimize}_{R^1, Z, L} \quad E(R^1, Z, L) + \theta_r^1 R^1 \\
g_2(O^1, A^1) & = \text{minimize}_{O^1, A^1} \quad E(O^1, A^1) + \theta_o O^1 + \theta_a A^1 \\
g_3(O^2, R^2) & = \text{minimize}_{O^2, R^2} \quad E(O^2, R^2) - \theta_o O^2 - \theta_r^1 R^2 \\
g_4(A^2, R^3) & = \text{minimize}_{A^2, R^3} \quad E(A^2, R^3) - \theta_a A^2 - \theta_r^2 R^3 \qquad (14)
\end{aligned}
$$

are the slave problems which are optimized separately and efficiently while treating the dual variables $\theta_r^1, \theta_r^2, \theta_o, \theta_a$ constant, and the master problem then optimizes these dual variables to enforce consistency. Next, we solve each of the sub-problems and the master problem.

**Solving subproblem** $g_1(R^1, Z, L)$**:**  Solving the sub-problem $g_1(R^1, Z, L)$ requires optimizing with only continuous variables $(R^1, Z, L)$. We follow a multi-scale LBFGS strategy [2] to optimize this part. Each step of the LBFGS approach requires evaluating the gradient of $g_1(R^1, Z, L)$ wrt. $R^1, Z, L$.

**Solving subproblem** $g_2(O^1, A^1)$**:**  The second sub-problem $g_2(O^1, A^1)$ involves only discrete variables $(O^1, A^1)$. The dual variable dependent terms add $\theta_o O^1$ to the object unary potential $\psi_i(O^1)$ and $\theta_a A^1$ to the attribute unary potential $\psi_i(A^1)$. Let $\psi'(O^1)$ and $\psi'(A^1)$ be the updated object and attribute unary potentials. We follow a filter-based mean-field strategy [11, 12] for the optimization. In the mean-field framework, given the true distribution $P = \frac{\exp(-g_2(O^1, A^1))}{\bar{Z}}$, we find an approximate distribution $Q$, where approximation is measured in terms of the KL-divergence between the $P$ and $Q$ distributions. Here $\bar{Z}$ is the normalizing constant. Based on the model in Sec. 2.2, $Q$ takes the form as $Q_i(O_i^1, A_i^1) = Q_i^{\mathcal{O}}(O_i^1) \prod_m Q_{i,m}^{\mathcal{A}}(A_{im}^1)$, where $Q_i^{\mathcal{O}}$ is a multi-class distribution over the object variable, and $Q_{i,m}^{\mathcal{A}}$ is a binary distribution over $\{0,1\}$. With this, the mean-field updates for the object variables take the following form:

$$Q_i^{\mathcal{O}}(O_i^1 = l) = \frac{1}{Z_i^O} \exp\{-\psi_i'(O_i^1) - \sum_{l' \in 1..K} \sum_{j \neq i} Q_j^{\mathcal{O}}(O_j^1 = l')(\psi_{ij}(O_i^1, O_j^1))\} \qquad (15)$$

where $\psi_{ij}$ is a potts term modulated by a contrast sensitive pairwise cost defined by a mixture of Gaussian kernels [12], and $Z_i^{\mathcal{O}}$ is per-pixel normalization factor. Given this form of the pairwise terms, as in [12], we can efficiently evaluate the pairwise summations in Eq. 15 using $K$ Gaussian convolutions. The updates for the attribute variables also take similar form (refer to the supplementary material).

**Solving subproblems** $g_3(O_2, R_2)$**,** $g_4(A_2, R_3)$**:**  These two problems take the following forms:

$$g_3(O_2, R_2) = \text{minimize}_{O_2, R_2} \sum_{c \in \mathcal{S}} \pi_{o^2}^c \psi(R_c^2) + \sum_{c \in \mathcal{S}} \pi_{r^2}^c \psi(O_c^2) - \theta_o O^2 - \theta_r^1 R^2 \qquad (16)$$

$$g_4(A^2, R^3) = \text{minimize}_{A^2, R^3} \sum_m \left( \sum_{c \in \mathcal{S}} \pi_{a^2, m}^c \psi(R_c^3) + \sum_{c \in \mathcal{S}} \pi_{r^3}^c \psi(A_c^{2,m}) \right) - \theta_a A^2 - \theta_r^2 R^3$$

Solving of these two sub-problems requires optimization with both the continuous $R^2$ and discrete $O^2, A^2$ variables respectively. However since these two sub-problems consist of higher order terms (described in Eq. 8) and dual variable dependent terms, we follow a simple co-ordinate descent strategy to update the reflectance and the object (and attribute) variables iteratively. The optimization of the object (and attribute) variables are performed in a mean-field framework, and a gradient descent based approach is used for the reflectance variables.

**Solving master problem** The master problem then updates the dual-variables $\theta_r^1, \theta_r^2, \theta_o, \theta_a$ given the current solution from the slaves. Here we provide the update equations for $\theta_r^1$; the updates for the other dual variables take similar form. The master calculates the gradient of the problem $E(R, Z, L, O, A|I)$ wrt. $\theta_r^1$, and then iteratively updates the values of $\theta_r^1$ as:

$$\theta_r^1 = \theta_r^1 + \alpha_r^1 \left( g_1^{\theta_r^1}(R^1, Z, L) + g_3^{\theta_r^1}(O^2, R^2) \right) \tag{17}$$

where $\alpha_r^t$ is the step size $t^{th}$ iteration and $g_1^{\theta_r^1}, g_3^{\theta_r^1}$ are the gradients w.r.t. to the $\theta_r^1$. It should be noted that we do not guarantee the convergence of our approach since the subproblems $g_1(.)$ and $g_2(.)$ are solved approximately. Further details on our inference techniques are provided in the supplementary material.

**Learning:** In the model described above, there are many parameters joining each of these terms. We use a cross-validation strategy to estimate these parameters in a sequential manner and thus ensuring efficient strategy to estimate a good set of parameters. The unary potentials for the objects and attributes are learnt using a modified TextonBoost model of Ladicky et.al. [9] which uses a colour, histogram of oriented gradient (HOG), and location features.

# 5    Experiments

We demonstrate our joint estimation approach on both the per-pixel object and attribute labelling tasks, and estimation of the intrinsic properties of the images. For the object and attribute labelling tasks, we conduct experiments on the NYU 2 [25] and Pascal [26] datasets both quantitatively and qualitatively. To this end, we annotate the NYU 2 and the Pascal datasets with per-pixel attribute labels. As a baseline, we compare our joint estimation approach against the mean-field based method [12], and the graph-cuts based $\alpha$-expansion method [9]. We assess the accuracy in terms of the overall percentage of the pixels correctly labelled, and the intersection/union score per class (defined in terms of the true/false positives/negatives for a given class as TP/(TP+FP+FN)). Additionally we also evaluate our approach in estimating better intrinsic properties of the images though qualitatively only, since it is extremely difficult to generate the ground truths for the intrinsic properties, e.g. reflectance, depth and illumination for any general image. We compare our intrinsic properties results against the model of Barron and Malik[2][2, 4], Gehler et.al. [5] and the Retinex model [17]. Further, only visually we also show how our approach is able to recover better smooth and de-noised depth maps compared to the raw depth provided by the Kinect [25]. In all these cases, we use the code provided by the authors for the AHCRF [9], mean-field approach [11, 12]. Details of all the experiments are provided below.

## 5.1    aNYU 2 dataset

We first conduct experiment on aNYU 2 RGBD dataset, an extended version of the indoor NYU 2 dataset [25]. The dataset consists of 725 training images, 100 validation and 624 test images. Further, the dataset consists of per-pixel object and attribute labels (see Fig. 1 and 3 for per-pixel attribute labels). We select 15 object and 8 attribute classes that have sufficient number of instances to train the unary classifier responses. The object labels corresponds to some indoor object classes as *floor, wall, ..* and attribute labels corresponds to material properties of the objects as *wooden, painted, ....* Further, since this dataset has depth from the Kinect depths, we use them to initialize the depth maps $Z$ for both our joint estimation approach and the Barron and Malik models [2-4].

We show quantitative and qualitative results in Tab. 1 and Fig. 3 respectively. As shown, our joint approach achieves an improvement of almost 2.3% , and 1.2% in the overall accuracy and average intersection-union (I/U) score over the model of AHCRF [9], and almost 1.5 % improvement in the

| Algorithm | Av. I/U | Oveall(% corr) |
|---|---|---|
| AHCRF [9] | 28.88 | 51.06 |
| DenseCRF [12] | 29.66 | 50.70 |
| Ours (OA+Intr) | 30.14 | 52.23 |

(a) Object Accuracy

| Algorithm | Av. I/U | Oveall(% corr) |
|---|---|---|
| AHCRF [9] | 21.9 | 40.7 |
| DenseCRF [12] | 22.02 | 37.6 |
| Ours (OA+Intr) | 24.175 | 39.25 |

(b) Attribute Accuracy

Table 1: *Quantitative results on aNYU 2 dataset for both the object segmentation (a), and attributes segmentation (b) tasks. The table compares performance of our approach (last line) against three baselines. The importance of our joint estimation for intrinsic images, objects and attributes is confirmed by the better performance of our algorithm compared to the graph-cuts based (AHCRF) method [9] and mean-field based approach [12] for both the tasks. Here intersection vs. union (I/U) is defined as $\frac{TP}{TP+FN+FP}$ and '% corr' as the total proportional of correctly labelled pixels.*

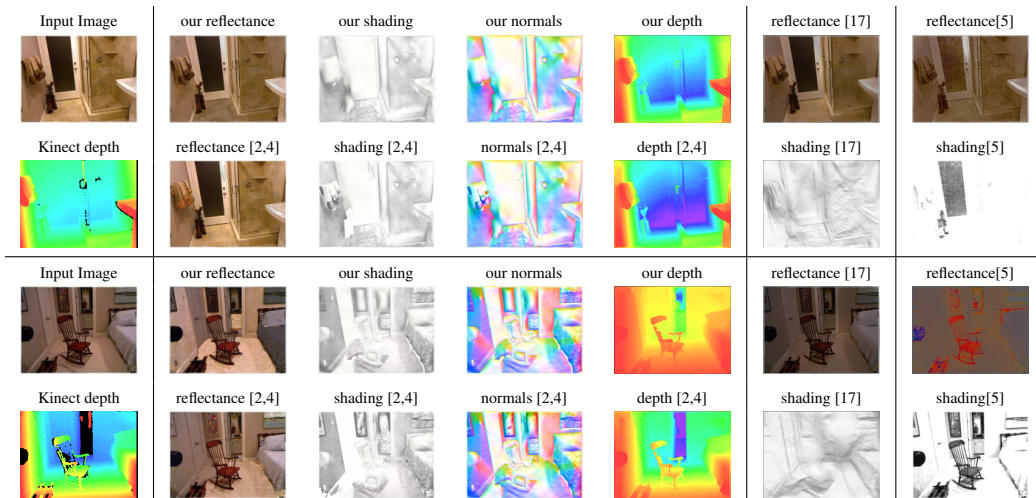

Figure 2: *Given an image and its depth image for the aNYU dataset, these figures qualitatively compare our algorithm in jointly estimating better the intrinsic properties such as reflectance, shading, normals and depth maps. We compare against the model Barron and Malik [2,4], the Retinex model [17] (2nd last column) and the Gehler et.al. approach [5] (last column).*

average I/U over the model of [12] for the object class segmentation . Similarly we also observe an improvement of almost 2.2 % and 0.5 % in the overall accuracy and I/U score over AHCRF [12], and almost 2.1 % and 1.6 % in the overall accuracy and average I/U over the model of [12] for the per-pixel attribute labelling task. These quantitative improvement suggests that our model is able to improve the object and attribute labelling using the intrinsic properties information. Qualitatively also we observe an improvement in the output of both the object and attribute segmentation tasks as shown in Fig. 3.

Further, we show the qualitative improvement in the results of the intrinsic properties in the Fig. 2. As shown our joint approach helps to recover better depth map compared to the noisy kinect depth maps; justifying the unification of reconstruction and objects and attributes based recognition tasks. Further, our reflectance and shading images visually look much better than the models of Retinex [17] and Gehler et.al. [5], and similar to the Barron and Malik approach [2,4].

## 5.2 aPascal dataset

We also show experiments on aPascal dataset, our extended Pascal dataset with per-pixel attribute labels. We select a subset of 517 images with the per-pixel object labels from the Pascal dataset and annotate it with 7 material attribute labels at the pixel level. These attributes correspond to *wooden, skin, metallic, glass, shiny...* etc. Further for the Pascal dataset we do not have any initial depth estimate. Thus, we start with a depth map where each point in the space is given same constant depth value.

Some quantitative and qualitative results are shown in Tab. 2 and Fig. 3 respectively. As shown, our approach achieves an improvement of almost 2.0 % and 0.5 % in the I/U score for the object and

| Algorithm | Av. I/U | Oveall(% corr) |
|---|---|---|
| AHCRF [9] | 32.53 | 82.30 |
| DenseCRF [12] | 36.9 | 79.4 |
| Ours (OA + Intr) | 38.1 | 81.4 |

(a) Object Accuracy

| Algorithm | Av. I/U | Oveall(% corr) |
|---|---|---|
| AHCRF [9] | 17.4 | 95.1 |
| DenseCRF [12] | 18.28 | 96.2 |
| Ours (OA+Intr) | 18.85 | 96.7 |

(b) Attribute Accuracy

Table 2: *Quantitative results on aPascal dataset for both the object segmentation (a), and attributes segmentation (b) tasks. The table compares performance of our approach (last line) against three baselines. The importance of our joint estimation for intrinsic images, objects and attributes is confirmed by the better performance of our algorithm compared to the graph-cuts based (AHCRF) method [9] and mean-field based approach [12] for both the tasks. Here intersection vs. union (I/U) is defined as $\frac{TP}{TP+FN+FP}$ and '% corr' as the total proportional of correctly labelled pixels.*

attribute labelling tasks respectively over the model of [12]. We observe qualitative improvement in the accuracy shown in Fig. 3.

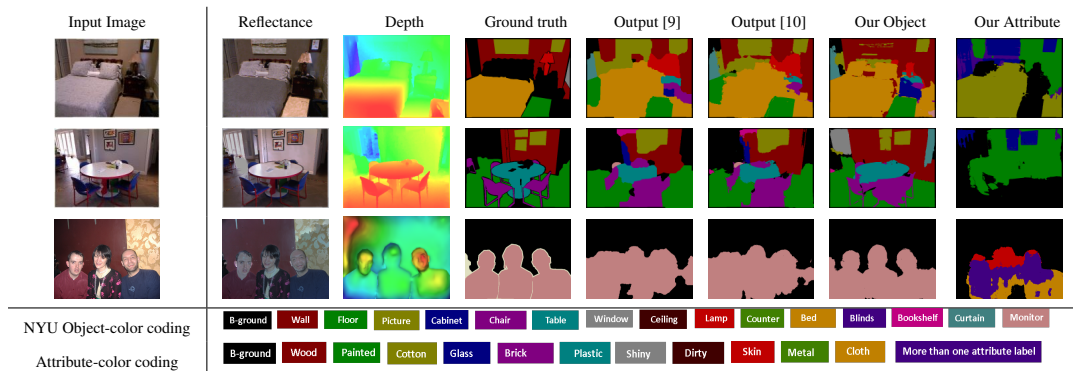

Figure 3: *Qualitative results on aNYU (first 2 lines) and aPascal (last line) dataset. From left to right: input image, reflectance, depth images, ground truth, output from [9] (AHCRF), output from [12], our output for the object segmentation. Last column shows our attribute segmentation output. (Attributes for NYU dataset: wood, painted, cotton, glass, brick, plastic, shiny, dirty; Attributes for Pascal dataset: skin, metal, plastic, wood, cloth, glass, shiny.)*

# 6 Discussion and Conclusion

In this work, we have explored the synergy effects between intrinsic properties of an images, and the objects and attributes present in the scene. We cast the problem in a joint energy minimization framework; thus our model is able to encode the strong correlations between intrinsic properties (*reflectance, shape,illumination*), objects (*table, tv-monitor*), and materials (*wooden, plastic*) in a given scene. We have shown that dual-decomposition based techniques can be effectively applied to perform optimization in the joint model. We demonstrated its applicability on the extended versions of the NYU and Pascal datasets. We achieve both the qualitative and quantitative improvements for the object and attribute labeling, and qualitative improvement for the intrinsic images estimation.

Future directions include further exploration of the possibilities of integrating priors based on the structural attributes such as *slanted, cylindrical* to the joint intrinsic properties, objects and attributes model. For instance, knowledge that the object is *slanted* would provide a prior for the depth and distribution of the surface normals. Further, the possibility of incorporating a mixture of illumination models to better model the illumination in a natural scene remains another future direction.

**Acknowledgements.** This work was supported by the IST Programme of the European Community, under the PASCAL2 Network of Excellence, IST-2007-216886. P.H.S. Torr is in receipt of Royal Society Wolfson Research Merit Award.

## Footnotes

[1]shading is the product of some shape and some illumination model which includes effects such as shadows, indirect lighting etc.

[2]We extended the SIRFS [2] model to our Scene-SIRFS using a mixture of reflectance and depth maps, and a single illumination model. These mixtures of reflectance and depth maps were embedded in the soft segmentation of the scene generated using the approach of Felzenswalb et.al. [21]. We call this model: Barron and Malik [2,4].

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
