[Reviews · NeurIPS 2013]

Submitted by Assigned_Reviewer_5

This paper is very exciting. The authors present a technique for jointly inferring the intrinsic properties of a scene (shapes, colors, illuminations) and a semantic-segmentation of a scene (objects labels, attributes). The technique basically works by gluing together Barron & Malik's SIRFS work with the DenseCRF work, where the "glue" holding the two models together is a dual-decomposition technique built around a segmentation technique. Doing these two tasks jointly seems to help both tasks --- not by a great deal, but probably enough.

The goal of this paper is great. Combining semantics and intrinsics is definitely something the computer vision community should focus on more, and I can see the machine learning community getting excited about the ideas here. The algorithm, though necessarily very complicated as it's a superset of complicated models, is presented well. The idea is novel, as is the overall model, though the machinery is not very novel itself.

I do have some concerns regarding clarity and evaluation. First, it's not clear to me how to contrast this model to Barron & Malik's stuff. Though the authors cite [4] (the scene-based version of the model) they don't seem to have used it. Instead they've made a new "scene" generalization out out what is presumably some past version of SIRFS. Can the authors clarify exactly what they did? It seems like the authors used the technique from [2] on different segments of the image, while sharing illumination across the entire image. This seems to be a technique for generalizing SIRFS to scenes, but it is not [4]. How does this technique work by itself for the task in [4], of just doing intrinsic reconstruction? Any why is only one illumination used? In [4] they get a lot out of modeling a mixture of lights, and many of the mistakes made by this model seem to be due to this incorrect assumption of a single global model of illumination. This needs to be explained well, because the author's "scene" version of [2] is used as a baseline technique by this paper, when it seems to be a novel algorithm of its own. Why not actually benchmark against [4]? The output here looks significantly worse than that of [4].

One completely unmentioned and very worrisome aspect of this paper is that in the NYU experiments seem to use a Kinect depth map as input. I assume this is true just because the output depth maps look too good to have been produced blind, by either this paper or by SIRFS. How is the Kinect being used? And what is being done in the PASCAL experiments in which there is not input depth? What sort of shape reconstructions are produced on those images? Some discussion of this is very necessary.

I'm confused by the comparison to [10]. The improvement of this paper with respect to [10] seems somewhat small. I appreciate the author's honesty in submitting a comparison to what is presumably another work of theirs, but why was this done? Are components of this model shared between these two works? If so, what are they? And does [10] use extra "verbal" information, as the attached paper suggests? Thankfully, the comparison against [12] seems correct, and this paper does improve the results by a medium-sized amount, which is enough to convince me that this technique works.

My greatest concern is that the improvement of this paper over [12] may be entirely due to the use of the Kinect depth input. It's possible that SIRFS is serving to do nothing except indirectly inform the attribute/object CRF components of the shape of the scene, which may be the only driving force behind the improved numbers. To verify that this is not the case, a simple baseline in which the DenseCRF also has access to the Kinect depth map (in some simple featurized form, of which several now exist) should also be presented.

Overall, I think this paper deserves credit for attempting an extremely ambitious task. The qualitative improvement (looking at the intrinsic output) is not very large, and the quantitative semantic improvement looks okay, but not very significant. I would really like my concerns regarding evaluation and the Kinect input to be addressed, as I feel like this will completely determine how I feel about the true contribution of this paper.
Summary: A very cool, ambitious paper about trying to solve all of computer vision. However, it has some potentially serious issues regarding evaluation and clarity with respect to experiments and past work.

Submitted by Assigned_Reviewer_6

This paper presents a model which estimates both intrinsic properties of images as well as object and attribute labeling. The intrinsic image labeling component comes from the SIRFs work of Barron et al, and the object/attribute labeling component comes from [10]. A probabilistic framework is proposed, and a dual decomposition strategy is used during function minimization. Experiments are shown on both NYU RGB-D and Pascal datasets, showing an improvement over several baselines.

Despite having many mathematical details, the paper is easy to read, well-motivated, and will likely excite the object recognition community. Showing that intrinsic image decomposition helps object class labeling is one of the strong points of the paper. While many of the ideas in this paper are not new, the combination of two separate lines of research make this paper exciting to read.
Summary: This paper combines two popular research avenues in computer vision, namely intrinsic image estimation and object/attribute labeling. The paper shows promising results, and introduces an easy-to-follow probabilistic framework using the language of modern machine learning.

Submitted by Assigned_Reviewer_7

The authors propose a technique for jointly estimating object class labels, attribute labels, as well as the intrinsic properties of an image, such as depth, illumination and reflectance. The model builds heavily on two previous works: [2] for estimating intrinsic properties, and [10] for estimating object and attribute labels. The proposed model extends [10] with new terms to accommodate for intrinsic properties, which are modelled as in [2]. Hence, it is a cross-over between the models in [2] and [10]. The authors' main motivation is to exploit synergistic effects that might arise between all these properties (e.g. estimating depth might help object labelling).


Quality and clarity:
the paper is rather hard to read because:
(a) it is very dense, with lots of definitions and equations following in close succession at a varying level of detail (e.g. some acronyms are not even spelled out, such as SIRFS; the learning paragraph at the end of sec. 4 is very vague);
(b) it lacks figures to illustrate the presented concepts;
(c) it is not self-contained, as part of the model are deferred to the supplementary material (e.g. section 2.2).


Originality:
Attempting to perform all the above estimation tasks at the same time is, in itself, rather original. However, the technique is incremental over [2,10].


Significance:
Given that essentially all model components are from [2,10] and that the proposed joint model does not perform much better than [10] on object and attribute labelling, this paper has limited significance.


Experiments:
On the positive side, the authors experiment on two reasonably sized datasets and compare to several previous works, including their base model [10]. On the negative side however, there are two important points:

- on the object and attribute labelling task, the proposed method performs barely better than [10]. Depending on dataset and performance measure, the 'improvement' is between +0.2% and +1.8%. This is quite weak and I wonder if we really need another paper to explain how this minor effect was obtained (i.e. by also jointly estimating intrinsic properties).

- even more importantly, the authors do not report quantitative evaluation for the other side of the story: estimating intrinsic properties such as depth, illumination and reflectance (e.g. by comparing to [2]). This is a major shortcoming of the experiments, as the main story of the paper is about the synergistic effects of estimating everything at the same time. The paper only evaluates whether intrinsic properties help labelling objects and attributes, but not the other way around.

All in all, the experiments do not convince of the alleged synergy, as the effect is very small in one direction, and not evaluated in the other direction.
Summary: Overall this is an ok paper which combines two rather mature lines of research: estimating object and attribute labels [10], and estimating intrinsic properties of the image, such as depth, illumination and reflectance [2]. Unfortunately, novelty is incremental, the paper is hard to read, and, importantly, the experiments do not support the main hope of the paper, i.e. to gain from the synergistic effects of estimating everything at the same time.
Author Feedback

Author rebuttal: We would like to thank the reviewers for their time and valuable comments on our paper, especially R5 and R6 for their positive responses. R5 and R6 acknowledge the novelty in the idea of exploring the synergy between the intrinsic properties and the objects and attributes present in a scene. We address the main concerns of R5 and R7 as follows.

R5: “it's not clear to me how to contrast this model to Barron & Malik's stuff. Though the authors cite [4] (the scene-based version of the model) they don't seem to have used it. Instead they've made a new "scene" generalization out what is presumably some past version of SIRFS.”

Ans: Note that we found out about [4] only some weeks before the NIPS paper submission. (CVPR camera ready was just over a month before NIPS submission deadline). There was no code provided by the authors’ of [4]. We extended the SIRFS [2] to our Scene-SIRFS using a mixture of reflectance and depth maps, and a single illumination model. These mixtures of reflectance and depth maps were embedded in the soft segmentation of the scene generated using the approach of Felzenswalb et.al. [21]. We had also initialized the Scene-SIRFS model with the Kinect depths on the NYU dataset. We call this model: “Barron and Malik [2,4]” see e.g. fig 4. We will make this point clearer in a final version. Finally, since the Scene-SIRFS approach is not limited to using a single global illumination model, we may also use a mixture of illumination models as done in [4] to further improve the robustness of our method, but this is our future direction to be explored (see line 431).

R5: “One completely unmentioned and very worrisome aspect of this paper is that in the NYU experiments seem to use a Kinect depth map as input. I assume this is true just because the output depth maps look too good to have been produced blind, by either this paper or by SIRFS. How is the Kinect being used? And what is being done in the PASCAL experiments in which there is not input depth? What sort of shape reconstructions are produced on those images? Some discussion of this is very necessary.”

Ans: Please see lines 318-319: “we use them to initialize the depth maps Z for our joint estimation approach.” For the Pascal dataset we do not have any initial depth map. Thus, we start with a depth map where each point in the space is given same constant depth value. We also apologize that we did not include results for depth maps for the Pascal dataset. In a final version we will add the shapes for the results in fig 3. The shapes are certainly not as good as with NYU. However in some cases (around 30%) the results look qualitatively acceptable. We will make these points clearer in a final version.

R5: “I'm confused by the comparison to [10]…Are components of this model shared between these two works? If so, what are they? And does [10] use extra "verbal" information, as the attached paper suggests? Thankfully, the comparison against [12] seems correct, and this paper does improve the results by a medium-sized amount, which is enough to convince me that this technique works.”

Ans: Both the papers [10] and [12] predict per-pixel object and attribute labels based on the mean-field approach, however [10] consists of object-attribute and attribute-attribute correlation terms. The use of such correlations terms helped [10] to improve the objects and attributes prediction, see sec. 2.2, thus we have made it as our baseline. We use all parts of [10] apart from the extra “verbal” information. Note that our approach is completely automatic.

R5: “My greatest concern is that the improvement of this paper over [12] may be entirely due to the use of the Kinect depth input…A simple baseline in which the DenseCRF also has access to the Kinect depth map (in some simple featurized form, of which several now exist) should also be presented.”

Ans: First, note that we also get an improvement over [12] for the Pascal dataset where no depth map is used. Secondly, thank you for the suggestion. We will add this experiment to a final version of this work. We suspect that the following may happen. Besides RGB-based appearance features we will add geometric features based on the depth maps. Note that these trained models will be input to both [12] and ours. Thus, it is likely that it would boost the performance of both [12] and our model.

R7 is concerned about the quality and clarity of the text.

Ans: To improve the clarity in a final version, we plan to include a table of definitions and notations. Note that R6 says: “Despite having many mathematical details, the paper is easy to read, well-motivated”

R7: “the authors do not report quantitative evaluation for the other side of the story: estimating intrinsic properties such as depth, illumination and reflectance”

Ans: We are aware of this concern and tried our best. We could not produce quantitatively results since there is no ground truth available for the reflectance, depth and illumination on these datasets (see lines 302-304). However we have qualitatively compared these properties against the models of retinex[17], Gehler et.al. [5] and our extended Scene-SIRFS model based on Barron and Malik [2]. Note, in [2] results on the MIT-intrinsic dataset were presented which consists of ground truth for the intrinsic properties. However there are around 20 images for 16 object classes. This is not enough to train the generic object and attribute models.

R7: “-on the object and attribute labelling task, the proposed method performs barely better than [10].”

Ans: In the early days of computer vision people have speculated about the synergy effect of intrinsic image properties and object recognition. However, in the past not many papers have addressed this topic. We believe that now is the right time to do so. Note that R7 says that both lines of research are rather mature. We see our work as a first step towards this grand goal rather than the final step.